# Gut microbiota composition and tumor immune features in meningioma patients

Kai Yin,[1] Shuai Ma,[2] Jichao Yang,[3] Mengzhao Feng,[1] Yuchao Zuo,[1] Fang Wang[1]

**ABSTRACT** Meningiomas are prevalent intracranial tumors with poorly understood extraneural drivers. While the gut-brain axis influences neuro-oncogenesis, meningioma-specific gut microbiome alterations and their clinical implications remain uncharacterized. This study integrated 16S rRNA sequencing of fecal samples from 15 treatment-naïve WHO grade I meningioma patients (MPs) and 15 healthy controls (HCs) with immunohistochemical profiling of tumor immune infiltrates (MPO+ neutrophils, CD68+ macrophages, CD3+ T cells). Compared with HCs, MPs exhibited significantly reduced alpha diversity (Shannon index, $P = 0.026$) and distinct beta diversity (permutational multivariate analysis of variance, $P < 0.0009$). Taxonomic analysis revealed enrichment of *Proteobacteria* (28.82% vs. 2.46%, $P = 0.001$), specifically *Escherichia_Shigella* at the genus level (25.95% vs. 1.61%, $P = 0.008$), along with depletion of *Bacteroidaceae* and *Ruminococcaceae*. LEfSe identified *Escherichia_Shigella* as the top meningioma-enriched biomarker. In diagnostic modeling, *Escherichia_Shigella* achieved an area under the receiver operating characteristic curve of 95.11% (95% CI: 86.91%–100%) for WHO grade I meningioma detection. Critically, *Escherichia_Shigella* abundance positively correlated with intratumoral MPO+, CD68+, and CD3+ cell densities (all $P < 0.05$), whereas *Ruminococcaceae* showed inverse correlations. The 16S rRNA sequencing data are publicly available in the GSA database under accession number CRA027974. This study provides the first evidence of gut dysbiosis in grade I meningioma, characterized by *Escherichia_Shigella* dominance and depletion of immunomodulatory commensals. This signature correlates with increased immune infiltration and holds promise as a novel biomarker.

**IMPORTANCE** Detailed exploration of host–microbe interactions can be worthwhile. Gut dysbiosis has been implicated in neuroinflammation, blood–brain barrier disruption, and oncogenesis in multiple cancer types, including gliomas. However, the gut microbiota composition and metabolic characteristics in patients with meningioma have not been previously reported. To address these critical knowledge gaps, we conducted a case–control study integrating 16S rRNA sequencing, clinical phenotyping, and immunohistochemical profiling. Our study revealed significant alterations in the gut microbiota of MPs, characterized by reduced alpha diversity, enrichment of *Proteobacteria*, and depletion of beneficial taxa, including *Bacteroidaceae* and *Ruminococcaceae*. Critically, we identified *Escherichia_Shigella* as a potential diagnostic biomarker and demonstrated strong correlations between elevated *Escherichia_Shigella*/Enterobacteriaceae abundance and increased intratumoral infiltration of MPO+ neutrophils, CD68+ macrophages, and CD3+ T cells. These findings are the first evidence that gut microbiome dysbiosis is closely associated with meningioma inflammation.

**KEYWORDS** gut microbiota, meningioma, gut-brain axis, 16S rRNA sequencing

**Peer Reviewer** Djandan Tadum Arthur Vithran, Central South University, Changsha, China

Address correspondence to Fang Wang, wangfang9282@zzu.edu.cn, or Yuchao Zuo, zuoyuchao328@126.com.

The authors declare no conflict of interest.

See the funding table on p. 12.

Meningiomas are the most prevalent primary tumors of the central nervous system in adults, comprising approximately 40% of intracranial tumors (1). These tumors originate from the meningeal arachnoid layer and therefore belong to the group of intracranial extra-axial neoplasms (2). Notably, certain WHO grade I meningiomas are prone to recurrence, with long-term follow-up studies indicating a recurrence rate of approximately 20%–39% (3). Importantly, there is significant variability in individual responses to these tumors. The mechanisms underlying such inter-individual variations remain poorly understood. Although the tumor is primarily composed of neoplastic cells, immunohistochemical analyses have consistently revealed the presence of infiltrating inflammatory and normal residual/reactive cells, such as macrophages/microglial cells and lymphocytes, in meningioma tissue specimens (4, 5). Infiltrating inflammatory cells play a role in the pathogenesis of various tumors, where they may be associated with distinct clinical behaviors.

The gut microbiota plays important roles in many physiological processes such as host–pathogen defense, energy, immunity, and metabolism (6). A significant variation in the gut microbial composition has been demonstrated between healthy individuals and patients with various types of cancers (7). Specifically, alterations in microbial diversity and enrichment of pro-inflammatory taxa (e.g., *Enterobacteriaceae*) have been associated with tumor progression and modulated anti-tumor immunity (8, 9). Recent studies have shown that the gut-brain axis is a pivotal pathway linking the gastro-intestinal microbiome to neurological health and disease (10–12). Gut dysbiosis has been implicated in neuroinflammation, blood–brain barrier disruption, and oncogenesis across multiple cancer types, including gliomas (13, 14). However, the role of the gut microbiome in meningioma pathogenesis remains unclear. To date, no comprehensive study has characterized meningioma-associated gut microbial signatures or evaluated their potential as diagnostic biomarkers or mediators of intratumoral immunity.

To address these critical knowledge gaps, we conducted a case–control study integrating 16S rRNA sequencing, clinical phenotyping, and immunohistochemical profiling. We hypothesized that patients with WHO grade I meningioma (MPs) exhibit gut microbial configurations that are distinct from those of healthy controls (HCs) and that specific dysbiotic features correlate with altered tumor immune microenvironments. This study aimed to: (i) characterize differences in gut microbial diversity, composition, and taxonomic abundance between MPs and HCs; (ii) identify meningioma-enriched microbial biomarkers with diagnostic potential; and (iii) investigate correlations between key microbiota and immune cell infiltration (neutrophils, macrophages, and T cells) in meningioma tissue.

Our findings revealed significant gut dysbiosis in patients with meningioma, dominated by enrichment of *Escherichia_Shigella* and depletion of commensal taxa. This study also exhibited the diagnostic performance of microbial signatures. Critically, we demonstrated novel associations between meningioma-associated microbiota and intratumoral immune responses, suggesting a potential role for the gut-brain axis in meningioma immunobiology.

## RESULTS

### Clinical characteristics of subjects

In our study, a total of 53 new participants were initially recruited, including 27 HCs and 26 MPs. Among these, seven who declined to participate or could not be contacted because of incomplete or outdated contact information were excluded. Furthermore, 16 participants were excluded from the study due to their use of probiotics or antibiotics, history of gastrointestinal diseases, alcoholism, smoking, prior surgery, or the presence of WHO grade II or III meningioma. Ultimately, 15 patients with MPs and 15 HCs successfully completed the clinical and fecal assessments according to the Consolidated Standards of Reporting Trials (CONSORT) flow diagram (Fig. S1).

The average age of MPs was 43.93 ± 4.28 years, and that of HCs was 42.33 ± 4.85 years. The average body mass index (BMI) of the two groups was 25.96 ± 2.12 and 26.57

± 2.16, respectively. No significant differences in age, gender, or BMI were observed between the patients and the HCs (*P* = 0.719, *P* > 0.99, *P* = 0.259, respectively). The relevant clinical data for each individual are presented in Table 1.

## Alterations of gut microbial diversity and composition in MP

Rarefaction curves for ASVs were constructed and visualized to assess the adequacy of the sample size utilized for the analysis. The graph indicates that the present sample size was adequate to represent the genetic diversity within the microbial community (Fig. 1A). The analysis indicated that there were 15 unique ASVs in the MP group compared to 10 in the HC group (Fig. 1B). MPs exhibited a marked reduction in gut microbial alpha diversity relative to HCs, as reflected by a significantly decreased Shannon index (Fig. 1C, *P* = 0.026). A significant difference in overall gut microbiota structure between the MP and HC groups was detected by permutational multivariate analysis of variance (PERMANOVA) applied to Bray–Curtis distances (Fig. 1D, *P* < 0.0009, $R^2$ = 0.068).

At the phylum level, *Firmicutes*, *Bacteroidetes*, *Proteobacteria*, *Actinobacteriota,* and *Actinobacteria* were the five dominant phyla, with total abundances of the microbiomes of the HCs and MPs, respectively (Fig. 1E and F). *Proteobacteria* (28.82% vs 2.46%, *P* = 0.001) was more abundant in MPs than in HCs (Fig. 1H). At the family level, *Enterobacteriaceae*, *Bacteroidaceae*, *Lachnospiraceae*, *Ruminococcaceae*, and *Tannerellaceae* were five with the highest relative abundance in the microbiome of the MP group, whereas *Bacteroidaceae*, *Ruminococcaceae*, *Prevotellaceae*, *Lachnospiraceae*, and *Veillonellaceae* were the five most abundant in that of the HC group (Fig. 1E through G). *Enterobacteriaceae* (1.81% vs 0.5%, *P* = 0.008) was more abundant in MPs than in HCs, whereas Firmicutes (27.89% vs 1.76%, *P* = 0.001) and *Bacteroidaceae* (18.12% vs 29.28%, *P* = 0.004) were less abundant (Fig. 1H). The genera-level characterization was more complicated. *Escherichia-Shigella*, *Bacteroides*, *Parabacteroides*, *Ligilactobacillus*, and *Blautia* were the five genera with the highest relative abundance in the microbiome of the MP group, while *Bacteroides*, *Faecalibacterium*, *Prevotella_9*, *Dialister*, and *Listeria* were the five most abundant genera in HCs (Table S1). *Escherichia-Shigella* (25.95% vs 1.61%, *P* = 0.008) was more abundant in MPs than in HCs, whereas *Firmicutes* (27.89% vs 1.76%, *P* = 0.001) and *Alistipes* (0.37% vs 4.39%, *P* = 0.004) were less abundant (Fig. 1H). The detailed relative abundances of these bacteria are shown in Table S1.

## Identification of key pathogenic bacterial genera in MP

We set the linear discriminant analysis (LDA) threshold to four and conducted an LDA effect size (LEfSe) to identify the microbial species with significant differences between the two groups. The species evolutionary branching diagram and LDA value distribution histogram showed differences on taxa abundance between groups. *Gammaproteobacteria*, *Proteobacteria*, *Enterobacteriaceae*, *Enterobacterales,* and *Escherichia_Shigella* were more abundant in the MP group, whereas *Bacteroides_plebeius*, *Alistipes*, *Rikenellaceae*, *Bateroidia,* and *Bateroidota* were more abundant in the HCs group (Fig. 2A and B). The

**TABLE 1** Clinical characteristics of subjects[a]

|  | MP | HCs | *P* value |
|---|---|---|---|
| Male, n (%) | 8 (53.3%) | 9 (60.0%) | > 0.99 |
| Age, Mean ± SD | 43.93 ± 4.28 | 42.33 ± 4.85 | 0.35 |
| BMI, Mean ± SD | 25.96 ± 2.12 | 26.57 ± 2.16 | 0.44 |
| Midline Location | 53.3% Midline; 46.7% Non-Midline | / | / |
| Skull Base Location | 46.7% Skull base; 53.3% Non-Skull base | / | / |
| Grade | WHO I | / | / |
| Histology | 66.7%Meningothelial, 20.0% fibrous, and 13.3%transitional meningioma | / | / |

[a]MP: meningioma patients, HCs: healthy controls, SD: standard deviation, BMI: body mass index, / indicates Not applicable.

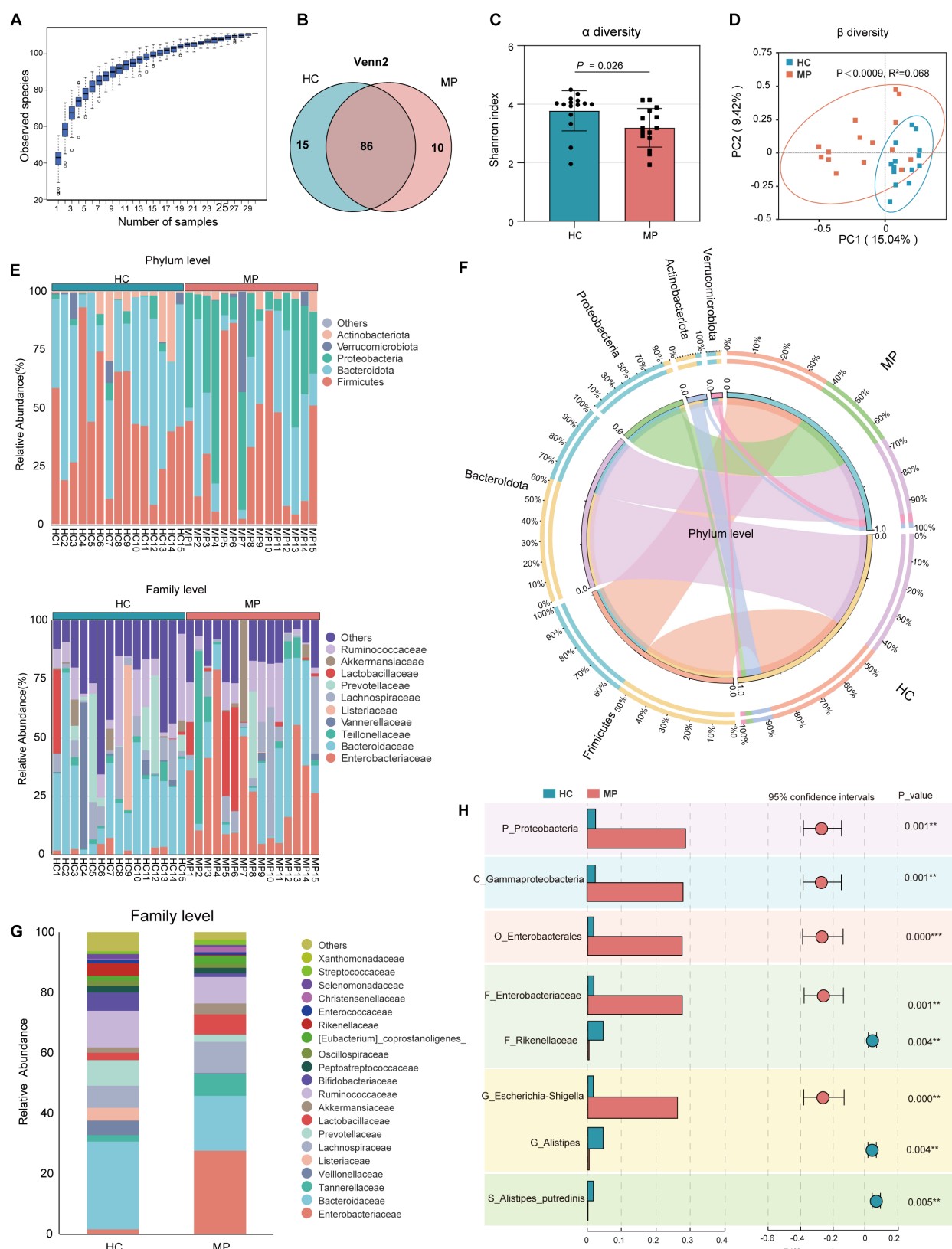

**FIG 1** Gut microbial community profiling in meningioma patients (MPs) and healthy controls (HCs). (A) Shannon-Wiener curves showing the relationship between numbers of fecal samples and estimated richness. (B) Venn diagram of shared and unique operational taxonomic units between the two groups. (C) Boxplots of alpha diversity as measured by the Shannon diversity. (D) Principal coordinate analysis of all samples based on weighted UniFrac distance. (Continued on next page)

(E) Relative distribution of bacterial composition at the phylum (up) and family (down) levels in MPs and HCs. (F) Circos diagram of sample-phylum correspondence. (G) Composition of the bacterial community (top 20) at the family level between HCs and MPs. (H) Statistical analysis of the gut microbiota in HCs and MPs groups by Mann-Whitney U test. Significance levels denoted in panels (C and H) represent adjusted P values.

overall results clearly indicated a significant difference in the gut microbes composition between the MP and the HC groups.

Subsequently, we evaluated the potential utility of the microbiota as a biomarker. To estimate prediction accuracy, we conducted a 10-fold cross-validation, partitioning the disease cohort, with 90% used as a training set and 10% used as a discovery set. A selection of the top 10 genera, including *Escherichia_Shigella*, which exhibited the largest effect size in the prediction model, was tested. This resulted in a mean area under the receiver operating characteristic curve (AUC) of 93.95% (Fig. 2C and D; 95% CI: 94.25%–100%). *Escherichia_Shigella* was frequently identified as the most significant genus, independently predicting MPs with an AUC of 95.11% (Fig. 2E, 95% CI: 86.91%–100%). Overall, our findings suggest that *Escherichia_Shigella* is one of the most promising biomarkers for the diagnosis of meningioma, having been identified as the optimal marker set.

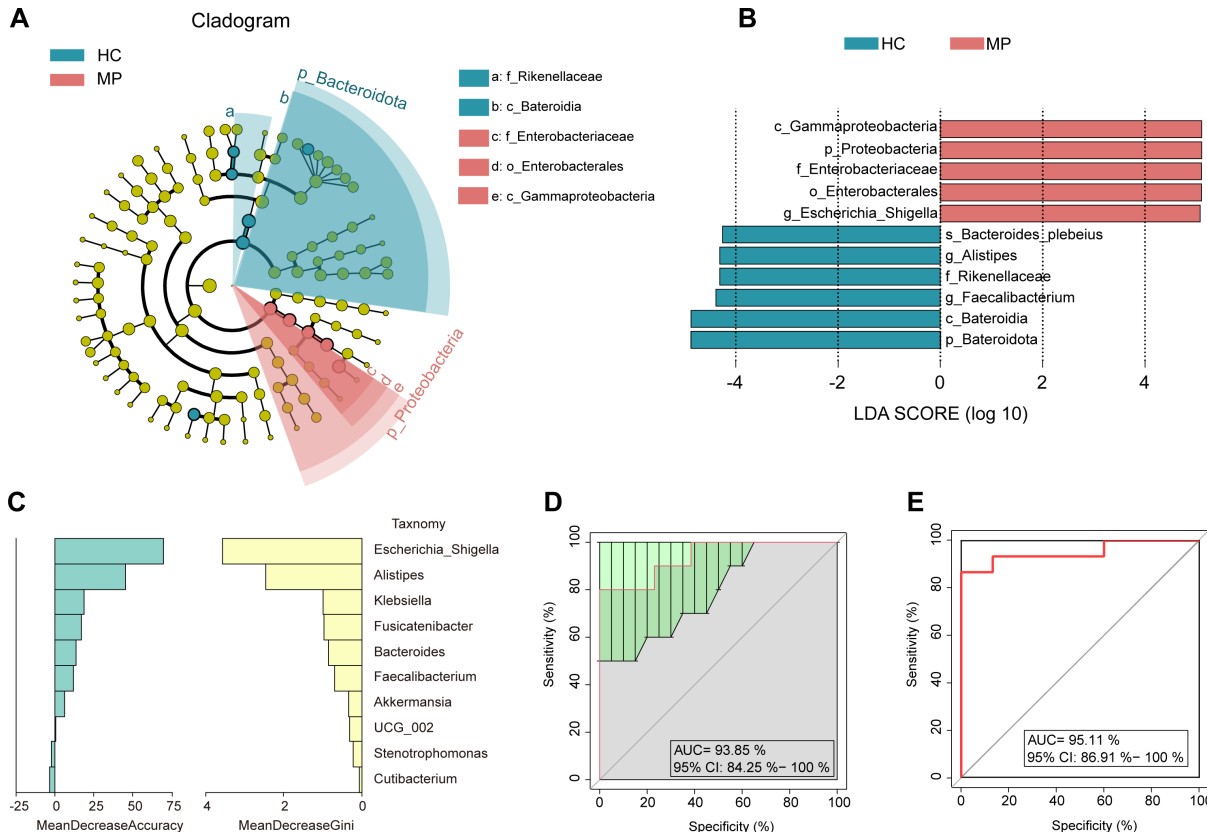

**FIG 2** Identification of differentially abundant taxa and diagnostic model evaluation. (A) Evolutionary cladogram depicting the phylogenetic distribution of gut microbial taxa differentially enriched between MP and HC groups (B) Linear discriminant analysis (LDA) score distribution plot ranking discriminative taxa by effect size, displaying only those meeting the predefined threshold (log10 LDA > 4.0). (C) A random forest classifier based on genus-level relative abundances identified ten differentially enriched taxa as the most discriminatory features between MP and HC groups. (D) The classification accuracy of a random forest model based on the ten most discriminative genera was evaluated and depicted as a receiver operating characteristic (ROC) curve. (E) The discriminatory power of *Escherichia_Shigella* abundance as a standalone biomarker was assessed by 10-fold cross-validation and visualized using a ROC curve.

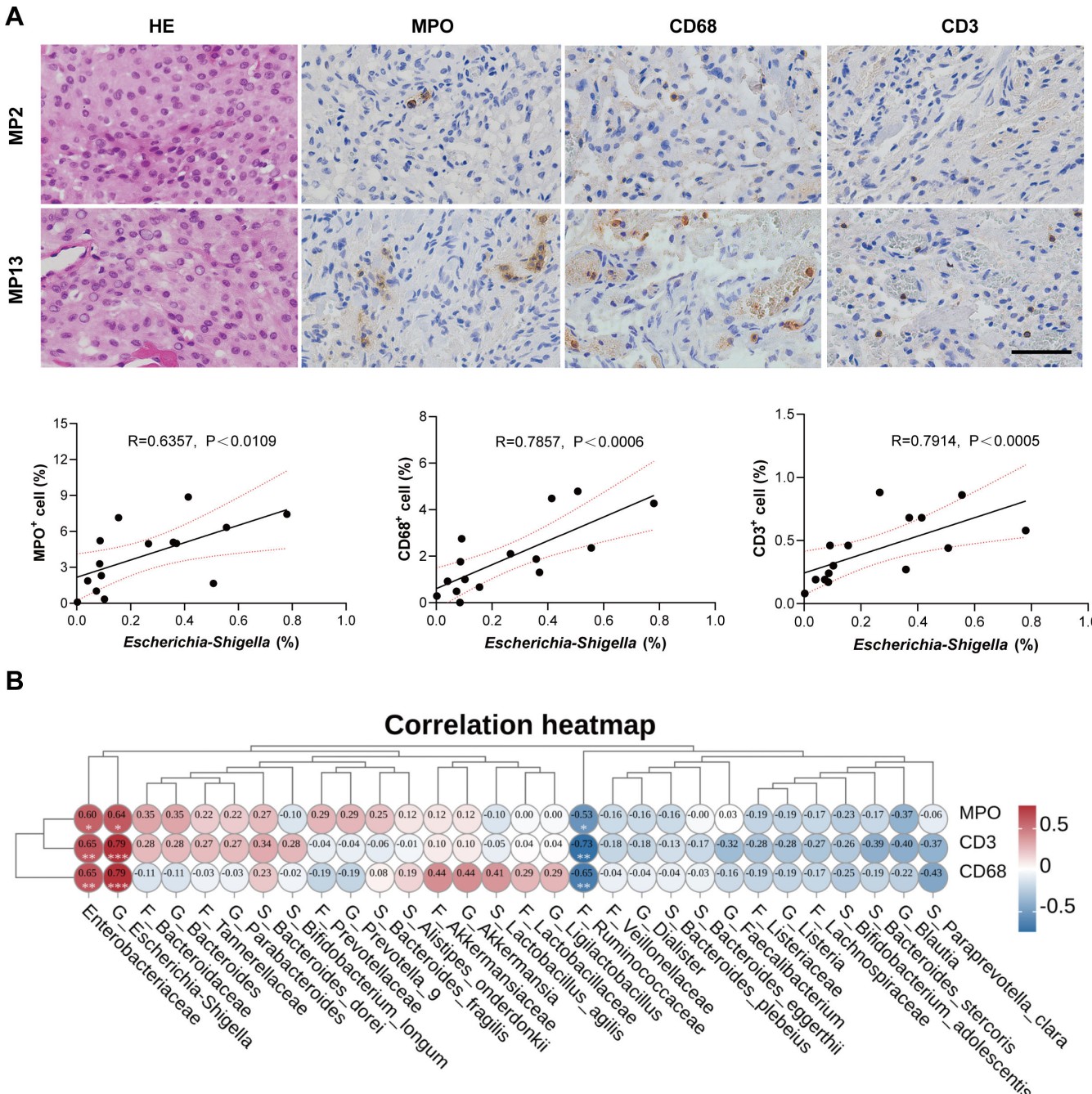

**FIG 3** Associations of specific gut bacteria with immune features in meningioma. (A) Representative HE and immunohistochemical images of MPO[+], CD3[+], and CD68[+] positive cells in meningioma tissues with low (MP2) versus high (MP13) *Escherichia_Shigella* relative abundance; scale bars, 50 µm. Scatter plots show correlations with immune cell counts. (B) Heatmap of Spearman's correlation between immune markers and differential taxa across taxonomic levels. Red, positive; blue, negative.

## Gut microbiota effects on meningioma immune response

Subsequently, we sought to ascertain whether alterations in the gut microbiome of patients with MPs were correlated with the immune response to meningioma. To investigate the impact of intestinal flora imbalance on meningioma immunity, tumor tissues were sectioned and subjected to immunohistochemical analysis for myeloperoxidase (MPO, indicative of neutrophils), CD68 (indicative of macrophages), and CD3 (indicative of T cells) staining. Relative expression levels of various immune cell types in

MP patients are shown in Fig. 3A. Spearman's correlation analysis revealed a significant positive association between the relative abundance of *Escherichia_Shigella* and the number of MPO[+], CD68[+], and CD3[+] cells (Fig. 3A). Additionally, a significant positive correlation was observed between the relative abundance of *Enterobacteriaceae* and the number of MPO[+], CD68[+], and CD3[+] cells. Conversely, the relative abundance of *Ruminococcaceae* was significantly negatively correlated with the populations of these immune cells (Fig. 3B). These findings suggest that alterations in the gut microbiota of patients with meningioma are linked to tumor immunity. Finally, we assessed the ratios of M1 to M2 macrophages (CD86[+]/Arg1[+]) and cytotoxic to regulatory T cells (CD8[+]/FoxP3[+]). Neither the M1/M2 nor the CD8[+]/FoxP3[+] ratio showed a significant correlation with *Escherichia_Shigella* abundance (Fig. S2). This indicates that the gut dysbiosis-associated increase in overall immune cell density is not accompanied by measurable polarization of these key functional subsets within the meningioma microenvironment.

## DISCUSSION

The application of technologies, such as high-performance sequencing, in molecular profiling and immunophenotyping of meningiomas may enable a more reliable prediction of biological behavior and subsequently have the potential to guide individualized meningioma therapy (15, 16). Significant differences between the gut microbial composition of healthy individuals and patients with cancer have been widely revealed by metagenomic analyses (17, 18). The gut microbiota plays important roles across the entire timeline of cancer, starting from its development and advancement, and ultimately also impacts the effectiveness of anti-tumor therapies (19). The microbiota-gut-brain axis is emerging as a rapidly growing field of study and a potential new therapeutic and early diagnostic target for central nervous system disorders (20). However, the composition and metabolic characteristics of the gut microbiota in patients with meningioma have not been previously reported.

Our study revealed significant alterations in the gut microbiota of MPs, characterized by reduced alpha diversity, enrichment of *Proteobacteria* (particularly *Escherichia_Shigella*), and depletion of beneficial taxa, including *Bacteroidaceae* and *Ruminococcaceae*. Critically, we identified *Escherichia_Shigella* as a potential diagnostic biomarker (AUC = 95.11%) and demonstrated strong correlations between elevated *Escherichia_Shigella*/*Enterobacteriaceae* abundance and increased intratumoral infiltration of MPO[+] neutrophils, CD68[+] macrophages, and CD3[+] T cells. These findings provide the first evidence of gut microbiome dysbiosis in meningioma and its association with the tumor immune microenvironment (Fig. 4).

The observations made in this study have interesting parallels with established gut-brain axis mechanisms in neuro-oncology; however, these also reveal meningioma-specific patterns (21). *Proteobacteria* expansion mirrors dysbiotic signatures in glioblastomas, where Enterobacteriaceae have been suggested to promote inflammation through LPS-mediated TLR4 activation (22, 23). However, the dramatic *Escherichia_Shigella* dominance (25.95% vs. 1.61% in HCs) exceeds reported levels in other CNS tumors, potentially pointing to distinct host–microbe interactions in meningioma. Conversely, the depletion of SCFA-producing *Ruminococcaceae* finds a counterpart in colorectal cancer, where loss of these commensals has been associated with impaired anti-tumor immunity (24).

The clinical implications of these findings are twofold. First, the exceptional diagnostic performance of *Escherichia_Shigella* (95.11% AUC) highlights its potential as a non-invasive biomarker for meningioma detection – a critical advance for a tumor lacking serum biomarkers. Second, the microbiota-immune correlations suggest that gut dysbiosis may actively shape the meningioma microenvironment. This suggests that the gut microbiome is a novel therapeutic target and that modulating these bacterial communities could potentially augment conventional treatments.

Based on these associations and insights from other pathological contexts, we propose a testable hypothesis linking the observed dysbiosis to meningioma

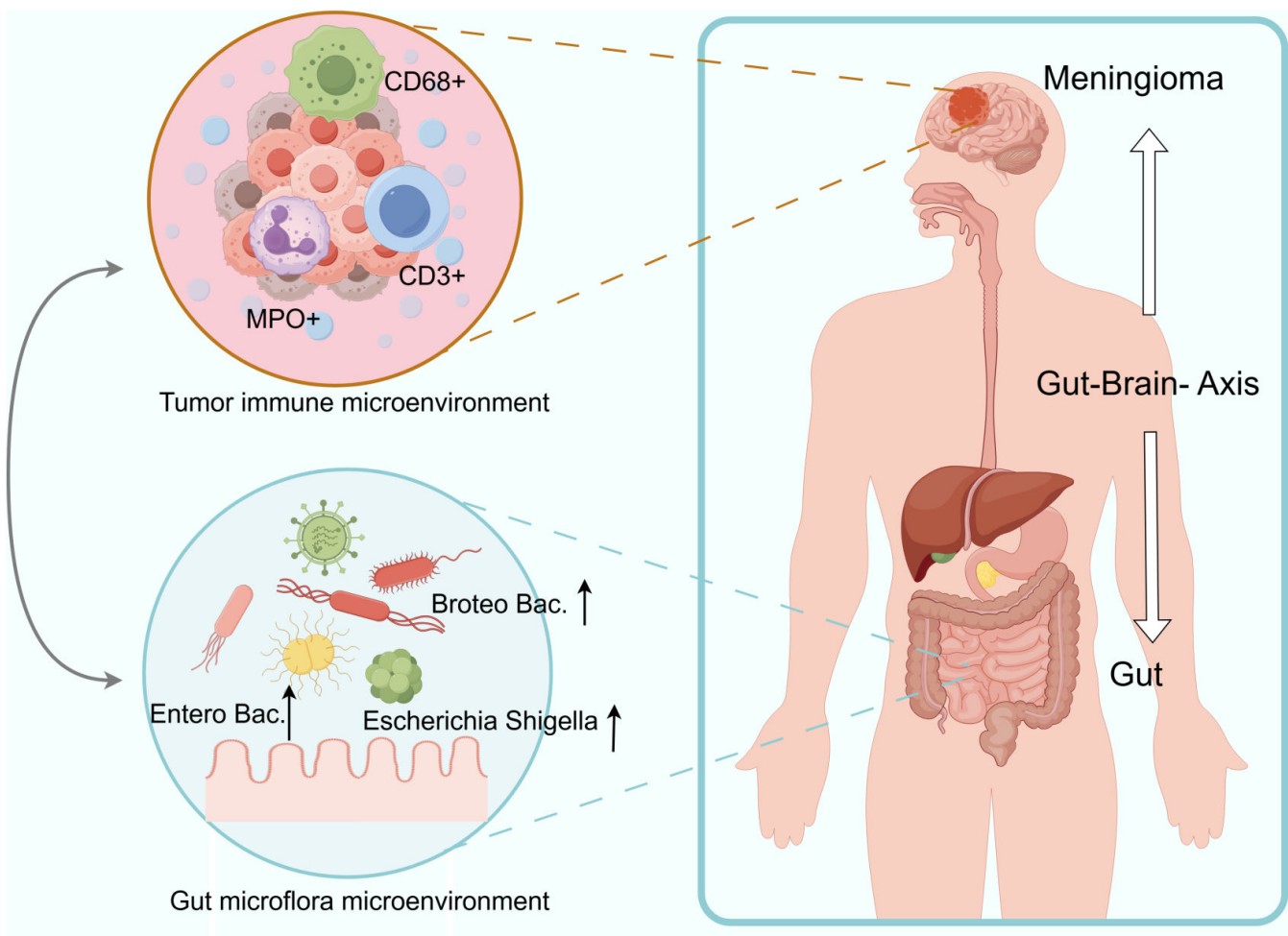

**FIG 4** Schematic diagram of the correlations between the meningioma immune microenvironment and intestinal flora. There are significant alterations in the gut microbiota of MPs, characterized by enrichment of *Proteobacteria* and strong correlations between elevated *Escherichia_Shigella/Enterobacteriaceae* abundance and increased intratumoral infiltration of MPO[+] neutrophils, CD68[+] macrophages, and CD3[+] T cells. Created by Figdraw.

immunobiology. Although this was not clearly demonstrated in the present study, several other studies have provided valuable insights. Gut dysbiosis particularly resulted in the enhancement of *Enterobacteriaceae* and *Akkermansiaceae,* which correlated with mastitis parameters. This is owing to LPS-mediated activation of TLR4/NF-κB signaling, promoting systemic inflammation (25). Outer membrane vesicles derived from *Escherichia coli* could induce the secretion of IL-8 and regulate neutrophil migration (26). Canale FP et al. reported that bacteria could increase intratumoral L-arginine concentrations, thereby increasing the number of tumor-infiltrating T cells (27). Concurrently, depletion of *Ruminococcaceae* – key producers of anti-inflammatory short-chain fatty acids such as butyrate – may compromise gut barrier integrity, thereby facilitating bacterial translocation and systemic immune dysregulation (28). These studies suggest a broader paradigm that may be relevant to meningiomas. However, these proposed mechanisms remain speculative in the context of meningiomas and require direct validation through future functional studies.

This study has several limitations that warrant consideration. First and most critically, the cross-sectional design precludes the determination of causality. We cannot discern whether the observed gut dysbiosis is a contributor to meningioma pathogenesis or a consequence of the tumor or associated systemic changes. Second, the modest sample size, which reduces statistical power and precludes meaningful subgroup analyses

by tumor grade or location. Additionally, 16S rRNA sequencing restricts functional insights; therefore, future studies should employ shotgun metagenomics to characterize microbial metabolic pathways (e.g., LPS biosynthesis) and host-interaction genes. Fourth, our findings are derived from a single-center cohort study. The lack of an external validation cohort indicates that the diagnostic performance and generalizability of the identified microbial signature require confirmation in larger multi-center studies. Finally, although we applied stringent exclusion criteria regarding medications and comorbidities, we cannot rule out the influence of unmeasured confounders—such as detailed dietary patterns, long-term geographic residency, and nuances of the hospital or home environment—on gut microbiota composition.

Based on these observations and their correlative nature, future research should prioritize the following aspects: longitudinal sampling in patients with meningioma (e.g., pre- and post-resection) to clarify the temporal dynamics and directionality of the gut–tumor relationship; functional and mechanistic validation using fecal microbiota transplantation in meningioma models to test whether *Escherichia_Shigella*-enriched microbiota can influence meningioma growth or the immune microenvironment; and multi-center, larger-scale validation of *Escherichia_Shigella* as a diagnostic biomarker across diverse cohorts. In such larger cohorts, employing multivariate and clustering approaches will be crucial to determine whether meningioma patients are characterized by distinct, stable microbial community structures, which may provide ecological insights and more robust biomarkers than single taxa. The therapeutic exploration of probiotics targeting *Ruminococcaceae* or selective antibiotics against Enterobacteriaceae represents a promising translational avenue. Integrating microbiome profiling with meningioma molecular subtypes could further personalize biomarker applications.

In this study, we present the first evidence linking gut dysbiosis to meningioma, characterized by *Escherichia_Shigella* dominance and depletion of immunomodulatory commensals. This microbial signature demonstrates high diagnostic accuracy and is associated with an increased density of tumor-infiltrating immune cells, implicating the gut microbiome as a novel biomarker source and potential therapeutic target.

## MATERIALS AND METHODS

### Clinical cohort

This study was approved by the Ethics Committees of the First Affiliated Hospital of Zhengzhou University and conducted in accordance with the Declaration of Helsinki (No. 2025-KY-0733-002). All participants provided written informed consent for the use of their samples. This study enrolled patients with newly diagnosed, treatment-naïve primary meningioma from the Department of Neurosurgery and recruited healthy people as controls from the Physical Examination Center at the First Affiliated Hospital of Zhengzhou University. The collected cases were diagnosed based on the tumor classification criteria of the WHO (29). Selection criteria were the following: (i) age >18 years and (ii) presence of WHO grade I meningioma on postoperative pathological examination. The exclusion criteria were as follows: (i) participants who had utilized probiotics, antibiotics, nonsteroidal anti-inflammatory drugs (NSAIDs), immunosuppressants, chemotherapy agents, or any other medications influencing the intestinal microbiota within one month prior to fecal sample collection; (ii) those who had undergone invasive procedures within seven days preceding fecal sample collection; (iii) individuals with a history of gastrointestinal diseases (including liver, gallbladder, pancreas, and spleen disorders), such as inflammatory bowel disease, irritable bowel syndrome, and chronic constipation, as well as those with a surgical history, excluding radical gastrectomy and exploratory laparotomy related to the current treatment; (iv) patients with a history of alcoholism, smoking, or substance abuse; (v) diagnosis of WHO grade II or III meningioma or any other concurrent malignant tumor; and (vi) those who declined to provide informed consent. The participant selection process, including patient exclusion reasons, is depicted in the accompanying flow diagram.

## Sample collection

Fecal samples were obtained via natural defecation. Prior to defecation, urine was evacuated to prevent contamination. A sampling spoon was used to transfer 2 grams of feces into a sterile fecal sample collection tube, which was then placed in an icebox to maintain a low-temperature environment and inhibit bacterial proliferation. The fecal samples were stored in a freezer at −80°C within two hours of collection.

Meningioma samples were obtained from patients undergoing surgical resection of primary meningioma tumors. During the resection procedure, specimens were immersed in normal saline and promptly placed on ice until further processing. The collected tissues were preserved in 4% paraformaldehyde for 48 h, then embedded in paraffin and sliced into 4-μm-thick sections for further analysis. Pathological diagnosis was performed by an expert neuropathologist at our institution, in accordance with the 2016 WHO classification of tumors of the central nervous system (29).

## DNA extraction and 16S ribosomal RNA gene sequencing

Microbial DNA was extracted from fecal samples using the cetyltrimethylammonium bromide and sodium dodecyl sulfate method, following the protocol provided by the manufacturer. The integrity and concentration of the extracted DNA were assessed via 1% agarose gel electrophoresis. The V3–V4 hypervariable regions of the 16S rRNA gene were amplified via polymerase chain reaction (PCR) employing specific primers (341F, 5′-CCTACGGGRSGCAGCAG-3′; 806R, 5′-GGACTACVVGGGTATCTAATC-3′) and barcodes. The PCR assays were conducted using a reaction mixture comprising 15 μL of Phusion High-Fidelity PCR Master Mix (New England Biolabs), 0.2 μM of each primer, and 10 ng of target DNA. The thermal cycling conditions were as follows: an initial denaturation at 98°C for 1 minute, followed by 30 cycles of denaturation at 98°C for 10 seconds, annealing at 50°C for 30 seconds, and extension at 72°C for 30 seconds, with a final extension step at 72°C for 5 minutes. After stringent quality filtering, denoising, and chimera removal, an average of 88,406 high-quality sequences per sample (range: 52,361–149,877) was obtained for downstream analysis, with average quality scores of Q20 > 98.6% and Q30 > 95.8% (detailed per-sample statistics are provided in Table S2). Purified DNA was sequenced on a NovaSeq platform, generating 250 base pair paired-end reads. The denoising process was executed using the DADA2 or Deblur module within the QIIME2 software (version QIIME2-202006) to derive initial ASVs. This DADA2 pipeline provided high-resolution ASVs for all downstream analyses. Taxonomic annotation was performed using the QIIME2 software with reference to the Silva database. To ensure the accuracy of the taxonomic assignment for key differentially abundant taxa, the representative ASV sequences identified as *Escherichia_Shigella* were extracted and subjected to a BLASTn search against the NCBI 16S ribosomal RNA sequence database (https://blast.ncbi.nlm.nih.gov/). This independent verification confirmed high sequence similarity (>99%) to established *Escherichia_Shigella* references, validating the annotation. Measures of alpha and beta diversity were calculated using QIIME2 and visualized employing the ggplot2 package in the R programming environment (http://www.R-project.org). The 16S rRNA gene amplicon sequencing data were obtained from fecal samples and provided by Beijing Novogene Biotechnology Co., Ltd, China.

## Histopathology analysis

For the hematoxylin-eosin (H&E) staining procedure, the slides were subjected to the standard H&E staining protocol to facilitate microscopic examination. Specifically, the slides were immersed in a hematoxylin solution (Solarbio, China) for 8 sec, followed by a 5-min rinse with distilled water. Subsequently, the sections were stained with an eosin solution (Solarbio, China) for 5 min and rinsed again with distilled water. The sections were then dehydrated using a series of graded alcohols and cleared in xylene. Finally, the stained sections were examined, and images were captured using an Olympus IX51 microscope (Olympus, Japan).

For immunohistochemical staining, the slides were washed in 0.5% PBS with Triton X-100 for 15 min, followed by incubation in PBS containing 0.5% bovine serum albumin for 0.5 h at room temperature. Then, the slides were incubated overnight at 4°C with the primary antibodies (anti-MPO: 66177-1-Ig, Proteintech; anti-CD68: ab125212, Abcam; and anti-CD3: 60181-1-Ig, Proteintech). After rinsing with PBS, the slides for immunohistochemistry were incubated with horseradish peroxidase-conjugated secondary antibodies for 1 h at room temperature and then stained with 3,3′-diaminobenzidine. Following dehydration in graded alcohol and clearing in xylene, the sections were observed and photographed using an Olympus IX51 microscope (Olympus, Japan).

## Statistical analysis

Statistical evaluation of clinical data was conducted with GraphPad Prism version 8.0. Values following a Gaussian distribution were reported as means ± standard deviation (SD) and were compared by Student's $t$-test; otherwise, the Mann-Whitney U test was adopted. Beta diversity was assessed via PERMANOVA with the adonis function from the vegan package in R. Visualization of sample dissimilarity was achieved by principal coordinate analysis via ggplot2 (R 4.0.2). Differentially abundant taxa were identified with the LEfSe pipeline. Correlation between microbiome features and clinical parameters was assessed with Spearman's rank-order coefficients. The diagnostic performance of *Escherichia_Shigella* abundance was evaluated using a random forest model (30). The AUC and its 95% CI were calculated via 10-fold cross-validation to provide a robust estimate and prevent overfitting.

## ACKNOWLEDGMENTS

This work was supported by the National Natural Science Foundation of China (No. 82503692), the Key Scientific and Technological Projects in Henan Province (No. 242102311220), the Key Scientific and Technological Projects in Henan Province (No. 242102310268), the Henan Provincial Department of Science and Technology Key Technology Development Project (No. 252102311181), and the Henan Provincial Health and Health Commission Joint Construction Project (No. LHGJ20230356).

All authors contributed to the study conception and design. K.Y., S.M., F.W., and Y.Z. were responsible for the initial assessing and diagnosing patients; J.Y. and M.F. were responsible for assessing and documenting their patients' health information; K.Y. and S.M. recorded and confirmed the data; analysis was performed by S.M. and K.Y.; and K.Y., F.W. wrote the final manuscript. All authors commented on previous versions of the manuscript. All authors read and approved the final manuscript.

Informed consent was obtained from all individual participants included in the study.

## AUTHOR AFFILIATIONS

[1]Department of Neurosurgery, The First Affiliated Hospital of Zhengzhou University, Zhengzhou, China
[2]Department of Neurosurgery, The Second Affiliated Hospital of Zhengzhou University, Zhengzhou, China
[3]Department of Neurology, Zhengzhou Central Hospital, Zhengzhou University, Zhengzhou, China

## AUTHOR ORCIDs

Yuchao Zuo 🔘 http://orcid.org/0000-0003-2876-9682
Fang Wang 🔘 http://orcid.org/0000-0002-1474-2297

## FUNDING

| Funder | Grant(s) | Author(s) |
| --- | --- | --- |
| National Natural Science Foundation of China | 82503692 | Fang Wang |
| Key Scientific and Technological Projects in Henan Province | 242102311220 | Fang Wang |
| Key Scientific and Technological Projects in Henan Province | 242102310268 | Yuchao Zuo |
| Henan Provincial Department of Science and Technology Key Technology Development Project | 252102311181 | Shuai Ma |
| Henan Provincial Health and Health Commission Joint Construction Project | LHGJ20230356 | Shuai Ma |

## AUTHOR CONTRIBUTIONS

Kai Yin, Data curation, Formal analysis, Investigation, Methodology, Resources, Software, Supervision, Writing – original draft, Writing – review and editing | Shuai Ma, Data curation, Funding acquisition, Methodology, Software, Writing – review and editing | Jichao Yang, Data curation, Formal analysis, Methodology, Software, Supervision, Writing – review and editing | Mengzhao Feng, Data curation, Formal analysis, Resources, Software, Writing – review and editing | Yuchao Zuo, Conceptualization, Funding acquisition, Software, Validation, Visualization, Writing – original draft, Writing – review and editing | Fang Wang, Conceptualization, Data curation, Funding acquisition, Supervision, Validation, Visualization, Writing – original draft, Writing – review and editing

## DATA AVAILABILITY

The raw sequence data reported in this study have been deposited in the Genome Sequence Archive (Genomics, Proteomics & Bioinformatics 2021) at the National Genomics Data Center (Nucleic Acids Res 2022), China National Center for Bioinformation/Beijing Institute of Genomics, Chinese Academy of Sciences (GSA: CRA027974).

## ETHICS APPROVAL

This study was performed in line with the principles of the Declaration of Helsinki. Approval was granted by the Ethics Committees of the First Affiliated Hospital of Zhengzhou University (No. 2025-KY-0733-002).

## ADDITIONAL FILES

The following material is available online.

### Supplemental Material

**Supplemental figures (Spectrum02485-25-s0001.docx).** Fig. S1 and S2.
**Table S1 (Spectrum02485-25-s0002.xlsx).** The relative abundance of intestinal microflora in all donors and patients.
**Table S2 (Spectrum02485-25-s0003.docx).** Summary of sequencing depth and quality metrics for all samples.

### Open Peer Review

**PEER REVIEW HISTORY (review-history.pdf).** An accounting of the reviewer comments and feedback.

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
