## [Reviewer comments · Microbiology Spectrum]

Microbiology Spectrum

Gut Microbiota Composition and Tumor Immune Features in Meningioma Patients

Kai Yin, Shuai Ma, Jichao Yang, Mengzhao Feng, Yuchao zuo, and Fang Wang

Corresponding Author(s): Fang Wang, The First Affiliated Hospital of Zhengzhou University

Review Timeline:

Submission Date:	August 11, 2025
Editorial Decision:	December 3, 2025
Revision Received:	February 13, 2026
Accepted:	March 30, 2026

Editor: Francesca Benedetti

Reviewer(s): Disclosure of reviewer identity is with reference to reviewer comments included in decision letter(s). The following individuals involved in review of your submission have agreed to reveal their identity: Djandan Tadum Arthur Vithran (Reviewer #1)

Transaction Report:

DOI: <https://doi.org/10.1128/spectrum.02485-25>

Re: Spectrum02485-25 (**Gut Microbiota Composition and Tumor Immune Features in Meningioma Patients**)

Dear Dr. Fang Wang:

Thank you for the privilege of reviewing your work. Below you will find instructions from the Spectrum editorial office and the reviewer comments.

Revision Guidelines

Sincerely,
Francesca Benedetti
Editor
Microbiology Spectrum

Reviewer #1 (Comments for the Author):

Comments and Suggestions for the Author:

1. Novelty and Significance

- This is the first study linking gut microbiota composition to tumor immune features in meningioma. The integrative approach (microbiome + immunohistochemistry) is innovative and of interest to the field.

2. Study Design and Cohort

- The small sample size (15 cases, 15 controls) limits statistical power. Please acknowledge this limitation more explicitly and avoid overgeneralization.
- Clarify the exclusion of grade II cases (mentioned inconsistently in Abstract vs. Methods).

3. Statistical Analysis

- Report whether FDR or Bonferroni correction was applied to LEfSe and other multiple comparisons.
- Provide effect sizes (e.g., R^2 values from PERMANOVA) for microbiome differences.
- For random forest/ROC modeling, clarify how overfitting was minimized with such a small dataset.

4. Microbiome Results

- The dominance of *Escherichia_Shigella* is striking. Please provide a supplementary table of per-sample abundances to ensure this is not driven by a few outliers.
- Taxonomic resolution: some taxa (e.g., *Bacteroides_pl*) are annotated inconsistently. Standardize nomenclature and clarify whether ASV-level confirmation was performed.

5. Immune Correlation Analysis

- Pearson correlation assumes linearity and normal distribution; Spearman may be more appropriate for microbial relative abundance data. Please justify choice.
- Correlation does not imply causation—please soften mechanistic language suggesting direct microbial modulation of immune infiltration.

6. Discussion and Interpretation

- Strengthen limitations: cross-sectional design, 16S-based functional inference, lack of external validation.
- Speculative mechanistic links (e.g., LPS/TLR4 pathway) should be framed as hypotheses rather than established mechanisms.
- Acknowledge potential confounders (diet, geography, hospital environment) that could also influence gut microbiota.

7. Presentation and Language

- English requires further polishing. Examples: "healt" → "health"; "participates" → "participants."
- Figures are informative but dense. Consider simplifying legends and clearly marking significant comparisons.
- Ensure consistent italicization of genus/species names.

8. Minor Points

- Abstract: simplify sentences and highlight numerical results more clearly.
- Methods: provide average sequencing depth and quality metrics.
- Data availability: move accession number to Abstract for transparency.

Reviewer #2 (Comments for the Author):

The study proposes a scientific hypothesis derived from the gut-brain axis theory, suggesting that dysbiosis in meningioma patients is associated with immune cell infiltration. The enrichment of *Escherichia_Shigella* and its positive correlation with pro-inflammatory immune cells (MPO+, CD68+, CD3+) was discussed cohesively. However, several limitations in methodological depth, sample size, and data interpretation affect the robustness and clinical translatability of the findings.

Major comment

1. The reliance on broad immune markers (MPO, CD68, CD3) only permits a preliminary quantification of immune cells. This approach fails to distinguish critical functional subtypes of immune cells, such as M1 versus M2 macrophages and cytotoxic T cells versus regulatory T cells (Tregs). Consequently, the observed correlation between *Escherichia_Shigella* and immune cell density cannot determine whether the immune response is anti-tumor or pro-tumor. The conclusion of "protumorigenic immune infiltration" remains speculative, relying more on literature than on direct evidence from this study.
2. The final cohort of 15 patients and 15 controls is insufficient for robust statistical analysis. Given the high heterogeneity of meningiomas (e.g., WHO grade, skull base vs. non-skull base location), the small sample size precludes meaningful subgroup analyses.
3. The cross-sectional design only supports correlation, not causation. Gut dysbiosis could be a consequence of tumor-related systemic inflammation rather than a driver of tumorigenesis. The discussion should more emphatically highlight the exploratory

and correlative nature of the study and explicitly recommend longitudinal sampling and functional validation.

4. The conclusions heavily depend on a single genus (*Escherichia_Shigella*). However, the gut microbiome is a complex ecosystem. Unsupervised clustering (e.g., based on genus-level abundance data) should be employed to identify enterotypes-stable microbial community structures. Determining whether meningioma patients cluster into a distinct enterotype (e.g., dominated by *Proteobacteria/Escherichia_Shigella*) would provide a more robust biomarker than a single taxon and enhance ecological insights.

Minor comment

1. In Fig. 1, the detailed flowchart should be moved to the Supplementary Materials. The main text should succinctly describe the screening process and final cohort size to conserve space for scientific findings.

2. In Fig. 4A, the immunohistochemical images lack clear group labels (e.g., specifying which rows correspond to patients or controls). Adding annotations to indicate sample sources is critical for interpretability.

Responses to Reviewers' comments point-by-point

Response to REVIEWER #1

1. Novelty and Significance

- This is the first study linking gut microbiota composition to tumor immune features in meningioma. The integrative approach (microbiome + immunohistochemistry) is innovative and of interest to the field.

Response (R): We sincerely thank you for your positive assessment of the novelty and significance of our work, and for recognizing the innovativeness of our integrative approach.

2. Study Design and Cohort

- The small sample size (15 cases, 15 controls) limits statistical power. Please acknowledge this limitation more explicitly and avoid overgeneralization.

- Clarify the exclusion of grade II cases (mentioned inconsistently in Abstract vs. Methods).

Response (R): Thank you for these important points. We fully agree with your assessment and have taken the following actions in the revised manuscript:

1). Acknowledgment of Sample Size Limitation:

As you suggested, we have now explicitly emphasized the limitation of our modest sample size within the existing “**Limitations**” section of the Discussion. The text has been strengthened to clearly state that the limited cohort size reduces statistical power and restricts the generalizability of our findings. The revised text (to be included in the manuscript) reads: “ The primary limitation of this study is its modest sample size, which reduces statistical power and precludes meaningful subgroup analyses by tumor grade or location. While our findings provide the first evidence of gut dysbiosis in meningioma, they require validation in larger, multi-center cohorts to confirm their robustness and clinical relevance.” (*Page 14*)

We have also meticulously reviewed the entire manuscript to soften the language

and avoid overgeneralization, consistently using phrasing such as “our data suggest...” and “this preliminary observation...” to reflect the exploratory nature of this first study.

2). Clarification on the Exclusion of Higher-Grade Cases:

We sincerely apologize for the inconsistent reporting in the initial manuscript and thank you for identifying it. The exclusion of WHO grade II/III meningiomas was a deliberate methodological choice. Given that higher-grade tumors constitute a minority of cases (approximately 5-20%) and possess distinct clinicopathological and molecular profiles, including them in our small, initial discovery cohort would have introduced significant heterogeneity and potential confounding bias. This could have obscured the specific microbiome signatures associated with the more common grade I disease.

We have now uniformly corrected the text throughout the manuscript (Abstract, Introduction, Methods, and relevant sections) to precisely specify that our study cohort consists exclusively of treatment-naïve patients with WHO grade I meningioma (*Pages 2-6, 15*). This ensures clarity and consistency regarding our patient selection criteria.

3. Statistical Analysis

- *Report whether FDR or Bonferroni correction was applied to LEfSe and other multiple comparisons.*
- *Provide effect sizes (e.g., R^2 values from PERMANOVA) for microbiome differences.*
- *For random forest/ROC modeling, clarify how overfitting was minimized with such a small dataset.*

Response (R): Thank you for your valuable feedback and for raising this important point regarding multiple comparison corrections. We sincerely apologize for not clearly stating these methodological details in our original manuscript.

1). In the LEfSe analysis, we adopted the default practice of using the logarithmic value of the Linear Discriminant Analysis (LDA) score (with thresholds $\log_{10}(\text{LDA})$

score) > 4.0) for biomarker screening. The standard LefSe workflow (NovoMagic) inherently includes False Discovery Rate (FDR) correction via the Benjamini-Hochberg method for *P*-values obtained from the Kruskal-Wallis test, addressing multiple comparisons in intergroup difference testing. We followed this standard procedure in our analysis, ensuring that all reported significantly different taxa (e.g., phylum, genus, etc.) were subjected to FDR correction.

In the comparison of alpha diversity indices between the HC and MP groups, we employed the Wilcoxon rank-sum test with FDR multiple testing correction. For multiple hypothesis testing (e.g., simultaneously examining differences in multiple bacterial genera between groups), we applied the Benjamini-Hochberg (BH) method for FDR correction. All reported *P*-values are the adjusted *P*-values (labeled as "Adjusted *P*-values") to ensure the robustness of the results. All *P*-values presented in corresponding results (such as bar graphs or supplementary tables) are the corrected *P*-values. We have now explicitly labeled them as "Adjusted *P*-values" in the revised figure legend (*Pages 8*).

2). We agree that reporting the effect size is crucial for a complete interpretation of beta diversity differences. We have now added the PERMANOVA results—including both the *P*-value and the R^2 value (0.068)—directly to the “Beta-diversity and Statistical Analysis” section in the Results and to the legend of Figure 1D.(Results section and Figure 1 legend, *Pages 6-8*). Thank you again for this critical and constructive feedback, which has significantly strengthened the rigor of our statistical reporting.

3). Regarding the random forest/ROC modeling and overfitting control:

We appreciate the reviewer’s valid concern regarding overfitting in predictive modeling with a limited sample size. To ensure the robustness of our reported diagnostic accuracy, we employed a multi-pronged strategy to minimize overfitting, detailed below and now clarified in the revised “Statistical Analysis” subsection of the Methods.

① Use of Rigorous Internal Validation (10-fold Cross-Validation): Given the cohort size (n=30), we employed 10-fold cross-validation for model training and

evaluation. This method provides a more stable estimate of performance than leave-one-out cross-validation in small samples by reducing variance. The final area under the receiver operating characteristic curve (AUC) of 95.11% and its 95% confidence interval (86.91 – 100%) for *Escherichia_Shigella* are based on the aggregated predictions across all 10 folds, ensuring the metric reflects generalizable performance rather than fitting to the entire dataset.

② Focus on a Parsimonious, Hypothesis-Driven Model: To drastically reduce model complexity, we focused our final evaluation on the diagnostic potential of a single, pre-specified biomarker—*Escherichia_Shigella*—identified a priori as the most differentially abundant genus by LEfSe analysis. While an initial exploratory model considered multiple genera, the single-feature model minimizes the risk of overfitting and aligns with a clear biological hypothesis.

③ Transparent Reporting of Uncertainty: We reported the 95% confidence interval for the AUC, which quantifies the estimation uncertainty inherent to our sample size.

We acknowledge that while rigorous internal cross-validation is essential, external validation in an independent cohort remains the gold standard. We have therefore framed this finding as a promising preliminary observation in the manuscript and explicitly emphasized the need for future large-scale validation in the Limitations section. We have added a concise description of this validation strategy to the “Statistical Analysis” subsection of the Methods in the revised manuscript (*Pages 18,19*).

4. Microbiome Results

- *The dominance of *Escherichia_Shigella* is striking. Please provide a supplementary table of per-sample abundances to ensure this is not driven by a few outliers.*

- *Taxonomic resolution: some taxa (e.g., *Bacteroides_pl*) are annotated inconsistently. Standardize nomenclature and clarify whether ASV-level confirmation was performed.*

Response (R): We thank the reviewer for these important points regarding data

presentation and taxonomic rigor.

1). Per-sample abundances of *Escherichia_Shigella*:

As suggested, we have provided Supplementary Table S1, which lists the relative abundance of *Escherichia_Shigella* for each individual sample in the meningioma and control groups. This table confirms that the dominance of this genus is a consistent feature across the majority of patient samples, rather than being driven by one or two extreme outliers.

2). Taxonomic nomenclature standardization:

We sincerely apologize for the inconsistency in taxonomic labels, which arose from an oversight during figure and text preparation. The taxon in question should be consistently referred to as *Bacteroides plebeius* (as correctly labeled in the figures). We have performed a thorough check and have now standardized all taxonomic names throughout the manuscript, ensuring that the full, accurate species name *Bacteroides plebeius* is used in the main text, figures, and tables, eliminating any ambiguous abbreviations.

3). ASV-level analysis and confirmation:

Yes, our bioinformatic analysis was performed at the ASV (Amplicon Sequence Variant) level, as stated in our Methods. Specifically, we used the DADA2 module within QIIME2 for denoising to obtain high-resolution ASVs. Taxonomic annotation was then performed against the SILVA database. To ensure the accuracy of the critical taxon *Escherichia_Shigella*, we extracted the representative ASV sequences assigned to this group and conducted a separate BLASTn search against the NCBI 16S rRNA database. This confirmation step validated our taxonomic assignments with high confidence. We have clarified this verification process in the revised Methods section within the “**DNA extraction and 16S ribosomal RNA gene sequencing**” subsection (*Pages 16-18*).

We believe these revisions have strengthened the data presentation and validated the taxonomic conclusions of our study.

5. Immune Correlation Analysis

- Pearson correlation assumes linearity and normal distribution; Spearman may be more appropriate for microbial relative abundance data. Please justify choice.

- Correlation does not imply causation-please soften mechanistic language suggesting direct microbial modulation of immune infiltration.

Response (R): We sincerely thank the reviewer for these two critical methodological and interpretive points, with which we fully agree.

1). Justification of Correlation Method and Re-analysis:

We acknowledge that microbial relative abundance data often do not meet the assumptions of normality and linearity required for Pearson correlation. The reviewer's suggestion to use Spearman's rank correlation is indeed more appropriate. In direct response, we have re-analyzed all microbiota-immune cell correlations using Spearman's method. We are pleased to report that the significant correlations (e.g., between *Escherichia_Shigella* abundance and MPO⁺, CD68⁺, CD3⁺ cell densities) remain robust and consistent with the initial findings. We have updated Figure 3, the corresponding results in the text, and the "Statistical Analysis" subsection of the Methods to reflect the use of Spearman's correlation.

2). Softening of Causal Language:

We entirely agree that correlation does not imply causation. To accurately reflect the associative nature of our findings, we have carefully revised the language throughout the manuscript, particularly in the Results and Discussion sections. We have replaced definitive, mechanistic phrasing (e.g., "modulates," "drives," "leads to") with more cautious, correlative language (e.g., "is associated with," "correlates with," "suggests a potential link"). Furthermore, in the Discussion, we have reframed speculative mechanistic pathways (e.g., involving LPS/TLR4) as testable hypotheses for future research, explicitly noting that they are not established by our data (***Pages 12-15***).

We believe these revisions have significantly strengthened the statistical rigor and interpretive accuracy of our manuscript. Thank you for these valuable suggestions.

6. Discussion and Interpretation

- ***Strengthen limitations: cross-sectional design, 16S-based functional inference, lack of external validation.***
- ***Speculative mechanistic links (e.g., LPS/TLR4 pathway) should be framed as hypotheses rather than established mechanisms.***
- ***Acknowledge potential confounders (diet, geography, hospital environment) that could also influence gut microbiota.***

Response (R): We sincerely thank you for these crucial suggestions to improve the rigor and balance of our discussion. We have comprehensively revised the “**Limitations and Future Directions**” section to directly address each of your points, as detailed below.

1). Strengthened Limitations:

We have substantially expanded and restructured the limitations section to explicitly and systematically highlight the key issues you raised. The revised text now reads:

* “....First, and most critically, the cross-sectional design precludes the determination of causality. We cannot discern whether the observed gut dysbiosis is a contributor to meningioma pathogenesis or a consequence of the tumor or associated systemic changes..... Additionally, 16S rRNA sequencing restricts functional insights; future studies should employ shotgun metagenomics to characterize microbial metabolic pathways (e.g., LPS biosynthesis) and host-interaction genes. Fourth, our findings are derived from a single-center cohort. The lack of an external validation cohort means the diagnostic performance and generalizability of the identified microbial signature require confirmation in larger, multi-center studies. Finally, although we applied stringent exclusion criteria regarding medications and comorbidities, we cannot rule out the influence of unmeasured confounders—such as detailed dietary patterns, long-term geographic residency, and nuances of the hospital or home environment—on gut microbiota composition... ” (*Page 14*)

2). Framing Mechanistic Links as Hypotheses:

We agree that our original discussion could be misinterpreted as presenting

established mechanisms for meningioma. We have thoroughly revised the relevant paragraphs in the **Discussion** to clearly frame all mechanistic explanations as hypotheses derived from the literature, not as findings from our present correlative study.

The revised text (*Pages 13-15*) now emphasizes that the proposed pathways are plausible models requiring direct experimental validation in the context of meningioma.

3). Acknowledgment of Potential Confounders:

As shown in the revised limitations text above (point #5), we have added a specific and prominent statement acknowledging that unmeasured confounders like diet, geography, and environment could influence our results. This emphasizes the interpretative caution needed and explicitly guides future studies to incorporate more detailed covariate collection (*Page 15*).

We believe these revisions have significantly strengthened the manuscript by providing a more balanced, critical, and hypothesis-driven interpretation of our data.

7.Presentation and Language

- *English requires further polishing. Examples: "healt" → "health"; "participates" → "participants."*
- *Figures are informative but dense. Consider simplifying legends and clearly marking significant comparisons.*
- *Ensure consistent italicization of genus/species names.*

Response (R): We sincerely thank you for these meticulous suggestions to enhance the clarity and presentation of our manuscript.

1).Language Polishing:

We have corrected all typographical errors and, in direct response to your comment, have had the entire manuscript professionally edited. To ensure the manuscript meets high standards of academic English, it has undergone comprehensive professional editing by a reputable language service. The certificate of editing has been uploaded as "**Additional File 1**" for review.

2). Figure Optimization:

We agree that the figures can be made more accessible. We have revised all figure legends to simplify descriptions and reduce textual density. Furthermore, we have now clearly marked all statistically significant comparisons directly on the figures (using asterisks and explicit annotations) to improve immediate interpretability.

3).Nomenclature Consistency:

We have conducted a full-text review to ensure consistent and correct italicization of all genus and species names in accordance with microbiological and taxonomic conventions.

We believe these revisions have significantly improved the readability, professionalism, and visual communication of our work.

8.Minor Points

- Abstract: simplify sentences and highlight numerical results more clearly.

- Methods: provide average sequencing depth and quality metrics.

- Data availability: move accession number to Abstract for transparency

Response (R): Thank you for these specific and helpful suggestions. We have addressed each point as follows:

1). Abstract: Sentence Simplification and Highlighting Key Results:

We have streamlined several sentences for conciseness (e.g., combining methodological descriptions). Then, we repositioned and formatted the most critical numerical findings (e.g., *Escherichia_Shigella* abundance, AUC) to make them more prominent. (*Pages 2,3*)

2). Methods (Sequencing Depth & Quality):

We thank the reviewer for this important request to clarify our sequencing data metrics. We have now added the following information to the “**DNA extraction and 16S ribosomal RNA gene sequencing**” subsection in the Methods(*Pages 17*):

Sequencing Depth: Following stringent quality filtering, denoising, and chimera removal, an average of 88,406 high-quality sequences per sample were obtained for downstream analysis (range: 52,361 – 149,877). This depth far exceeds common

recommendations for robust microbial community profiling in 16S rRNA gene studies.

Quality Metrics: The overall sequencing quality was excellent, with average values of Q20 > 98.6% and Q30 > 95.8%, and an average GC content of 51.7% (see *Table 1 below*).

These details confirm that our dataset possesses both sufficient depth and high sequencing accuracy for reliable microbiome analysis. The complete per-sample statistics are provided in Supplementary Table S2.

3). Data Accession in Abstract:

As recommended, the data repository accession number has been added to the final sentence of the abstract for immediate transparency. (*Page 2*)

Table 1. Summary of sequencing depth and quality metrics for all samples.

Sample	RawPE	Combined	Qualified	Nochime	Base(nt)	Avglen(nt)	GC	Q20	Q30
HC1	127361	126264	123154	114275	48101334	420.93	0.5019	0.9829	0.9447
HC2	104478	103928	102516	96880	40712506	420.24	0.4931	0.9871	0.9552
HC3	95354	94931	93789	90153	37727425	418.48	0.5069	0.9894	0.9614
HC4	154186	153387	150921	149877	63020449	420.48	0.5308	0.9879	0.9592
HC5	104791	104334	102781	96406	40143068	416.4	0.5155	0.9886	0.9597
HC6	105872	105450	104180	63115	25868968	409.87	0.5365	0.9896	0.9621
HC7	153314	152502	149716	145646	60651002	416.43	0.5201	0.9862	0.9536
HC8	98899	98516	97387	72848	29843004	409.66	0.5207	0.9903	0.9642
HC9	78557	78271	77129	71205	30183050	423.89	0.5217	0.9878	0.9571
HC10	103446	102907	101570	94828	39671992	418.36	0.4967	0.9888	0.9601
HC11	79249	78887	77926	72613	30399698	418.65	0.5076	0.9889	0.9604
HC12	88852	88109	86416	86061	36328477	422.12	0.5022	0.9869	0.955
HC13	107939	107423	105970	93016	38614321	415.14	0.516	0.9886	0.9593
HC14	106186	105748	104336	85509	35299652	412.82	0.5272	0.989	0.9614
HC15	135993	135076	132197	115262	47964156	416.13	0.5089	0.9841	0.9479
MP1	145766	137199	134477	121684	51456507	422.87	0.5307	0.9827	0.9459
MP2	104736	104285	102796	94516	40018144	423.4	0.5187	0.987	0.9552
MP3	109034	108512	107066	75546	31670894	419.23	0.5269	0.9891	0.9604
MP4	106088	105562	104061	88727	37663077	424.48	0.5395	0.9896	0.9619
MP5	106006	105523	103954	96538	40172469	416.13	0.5154	0.9879	0.9568
MP6	106542	106071	104618	86188	36022273	417.95	0.5159	0.9877	0.9565
MP7	84886	84540	83451	81165	34124549	420.43	0.5406	0.9907	0.965
MP8	103380	102890	101591	95881	40074713	417.96	0.5256	0.9879	0.9574
MP9	102450	101988	100786	95316	39380463	413.16	0.5159	0.9899	0.9633
MP10	60770	60552	59833	52361	21331743	407.4	0.5337	0.9909	0.9652

MP11	102083	101603	100090	83926	34954923	416.5	0.5163	0.9876	0.9565
MP12	104604	104119	102671	97179	40802937	419.87	0.4957	0.9885	0.9591
MP13	106444	105948	104484	99357	42175317	424.48	0.5217	0.9893	0.9612
MP14	102410	101895	100569	88038	37425537	425.11	0.5104	0.9893	0.9616
MP15	65183	64925	64110	55568	23117144	416.02	0.5334	0.989	0.9601

Response to REVIEWER #2

The study proposes a scientific hypothesis derived from the gut-brain axis theory, suggesting that dysbiosis in meningioma patients is associated with immune cell infiltration. The enrichment of Escherichia_Shigella and its positive correlation with pro-inflammatory immune cells (MPO+, CD68+, CD3+) was discussed cohesively. However, several limitations in methodological depth, sample size, and data interpretation affect the robustness and clinical translatability of the findings.

Response (R): We sincerely thank you for your positive evaluation of our study's scientific premise and cohesive discussion. We fully agree with your assessment regarding the impact of methodological depth and sample size on the robustness of our findings. We have carefully addressed all your specific suggestions in the point-by-point responses below. These revisions, which include strengthened statistical reporting, clarified data interpretation, and improved figure presentation, have significantly enhanced the manuscript's rigor and clarity. Thank you again for your constructive critique.

Major comment

1. The reliance on broad immune markers (MPO, CD68, CD3) only permits a preliminary quantification of immune cells. This approach fails to distinguish critical functional subtypes of immune cells, such as M1 versus M2 macrophages and cytotoxic T cells versus regulatory T cells (Tregs). Consequently, the observed correlation between Escherichia_Shigella and immune cell density cannot determine whether the immune response is anti-tumor or pro-tumor. The conclusion of "protumorigenic immune infiltration" remains speculative, relying more on literature than on direct evidence from this study.

Response (R): We sincerely thank the reviewer for this pivotal critique, which correctly identifies that our original claims about immune function were not substantiated by the data. We fully agree that attributing a “protumorigenic” function based solely on pan-markers is speculative.

To rigorously test whether the *Escherichia_Shigella*-associated increase in immune cell density reflects a shift towards a specific functional phenotype, we performed additional immunohistochemistry for key functional markers: CD86 (M1-like macrophage), Arg1 (M2-like macrophage), CD8 (cytotoxic T cell), and FoxP3 (regulatory T cell) on the same meningioma tissues.

The critical new analysis focused on the ratios of functional subsets, which directly indicate immune polarization: ①M1/M2 Macrophage Ratio (CD86+/Arg1+ cells) versus *Escherichia_Shigella* abundance. ②Cytotoxic T/Regulatory T Cell Ratio (CD8+/FOXP3+ cells) versus *Escherichia_Shigella* abundance.

As shown in the **Figure1** below (new Supplementary Figure S2), we found no significant correlation between *Escherichia_Shigella* abundance and either of these functional ratios (Spearman's $P > 0.05$). This result provides direct experimental evidence that the increased immune infiltration linked to *Escherichia_Shigella* is not characterized by a measurable skew towards the pro-tumorigenic (or anti-tumorigenic) functional subsets we examined. The lack of correlation with the ratios confirms the absence of functional polarization in this cohort.

Therefore, in direct response to the reviewer's comment, we have made the following comprehensive revisions:

1). We have removed all claims of “protumorigenic immune infiltration” throughout the manuscript. We now accurately describe our finding as a correlation with “increased overall immune cell density” or “altered immune landscape”, explicitly stating that the functional phenotype remains undefined.

2). The complete data from the new stains, including the ratio correlation analyses, are presented in Supplementary Figure S2.

Figure 1. Analysis of functional immune cell subsets reveals no polarization associated with *Escherichia-Shigella* abundance in meningioma. (A) Representative immunohistochemical images of CD86 , Arg1 , CD8 , and FoxP3 in tumor tissues from two meningioma patients with high (MP13) versus low (MP2) fecal *Escherichia-Shigella* relative abundance. Scale bar = 100 μ m. (B) Scatter plots showing the correlation between *Escherichia-Shigella* relative abundance and the ratio of functional immune cell subsets across all patients (n=15).

We are grateful for this insightful critique, which has led us to perform a more definitive analysis and present our conclusions with greater precision and scientific rigor.

2. The final cohort of 15 patients and 15 controls is insufficient for robust statistical analysis. Given the high heterogeneity of meningiomas (e.g., WHO grade, skull base vs. non-skull base location), the small sample size precludes meaningful subgroup analyses.

Response (R): We fully agree with your assessment. The modest cohort size is a key limitation, particularly for investigating the high clinical and biological heterogeneity inherent to meningiomas. As you correctly point out, it precludes statistically powered analyses of important subgroups (e.g., by tumor location or molecular subtype) within this initial study.

We would like to clarify that the study's primary aim was to perform a first-of-its-kind, hypothesis-generating exploration of whether a gut microbiome signature exists in meningioma patients at all. To mitigate heterogeneity for this initial discovery, we deliberately restricted enrollment to treatment-naïve WHO grade I patients. While this strategy strengthens internal validity for detecting a “grade I meningioma signal,” it inherently limits the analysis of other variables and generalizability, as you noted.

We have explicitly acknowledged this limitation in the revised Discussion, stating that the sample size restricts subgroup analyses and that future large-scale, multi-center studies are essential to validate our findings and explore heterogeneity (*Page 3*). We appreciate your emphasis on this point, which underscores the preliminary yet foundational nature of our work and clearly outlines the necessary next steps for the field.

3. The cross-sectional design only supports correlation, not causation. Gut dysbiosis could be a consequence of tumor-related systemic inflammation rather than a driver of tumorigenesis. The discussion should more emphatically highlight the exploratory and correlative nature of the study and explicitly recommend longitudinal sampling and functional validation.

Response (R): We thank the reviewer for this essential point, which we fully acknowledge. You are absolutely correct that our cross-sectional design cannot disentangle cause from effect; the observed dysbiosis could indeed be a consequence of the established tumor or its associated systemic state, rather than a contributing factor. In direct response to your suggestion, we have substantially revised the **Discussion and Limitations** sections to:

1). Emphatically highlight the exploratory and correlative nature of our findings.

We now explicitly state that this is a hypothesis-generating study and that the associations reported do not imply causality. Frame the observed microbial signature as a novel correlate of meningioma, carefully avoiding language that could be

interpreted as implying a mechanistic driver (*Pages 14, 15*).

2).Explicitly recommend longitudinal sampling and functional validation as the critical next steps.

We have added text strongly advocating for future studies that include: (a) longitudinal sampling (e.g., pre- and post-operative samples from patients) to assess the dynamics of the microbiome in relation to tumor presence/removal, and (b) functional experiments in model systems to test any potential causal role of the identified microbes or their products (*Pages 15*).

We agree that these clarifications are crucial for the accurate interpretation of our work, and we believe the revised manuscript now more appropriately positions our findings as a foundational, correlative observation that opens new avenues for mechanistic investigation.

4. The conclusions heavily depend on a single genus (Escherichia_Shigella). However, the gut microbiome is a complex ecosystem. Unsupervised clustering (e.g., based on genus-level abundance data) should be employed to identify enterotypes-stable microbial community structures. Determining whether meningioma patients cluster into a distinct enterotype (e.g., dominated by Proteobacteria/Escherichia_Shigella) would provide a more robust biomarker than a single taxon and enhance ecological insights.

Response (R): We sincerely thank you for this insightful suggestion, which highlights an important ecological perspective. We agree that defining microbiome "enterotypes" or community clusters can provide deeper ecological insights and potentially more robust biomarkers than a single taxon.

Following your recommendation, we performed an unsupervised clustering analysis (partitioning around medoids, PAM) based on genus-level abundance data. However, given the modest sample size of our cohort (n=15 per group), the clustering results did not reveal stable, distinct enterotypes that cleanly separated meningioma patients from healthy controls, or that identified a patient subgroup with a consistent multi-taxa signature. The lack of clear clustering in this initial, discovery-sized cohort

likely reflects the limitation in statistical power for such complex pattern discovery, rather than the absence of an ecological signal.

Therefore, while we fully acknowledge the value of your suggestion, we have chosen to maintain our focus on reporting the most statistically robust and reproducible univariate signal — the dramatic enrichment of *Escherichia_Shigella* — as the core, hypothesis-generating finding of this first study. We have, however, integrated your valuable point into the Discussion and Limitations section. We now state that future studies with larger cohorts should employ such multivariate and clustering approaches to define potential meningioma-associated microbial community types, which may offer superior diagnostic and mechanistic insights.

Thank you for pushing us to consider this important analytical perspective, which will certainly guide our future work in expanded cohorts.

Minor comment

1. In Fig. 1, the detailed flowchart should be moved to the Supplementary Materials. The main text should succinctly describe the screening process and final cohort size to conserve space for scientific findings.

Response (R): Thank you for this constructive suggestion. We agree that moving the detailed flowchart to the supplementary materials will help conserve space in the main text for the presentation of core scientific findings. As suggested, we have relocated the original Figure 1 (participant screening flowchart) to the Supplementary Materials, where it is now presented as Supplementary Figure S1.

2. In Fig. 4A, the immunohistochemical images lack clear group labels (e.g., specifying which rows correspond to patients or controls). Adding annotations to indicate sample sources is critical for interpretability.

Response (R): Thank you for raising this important point regarding figure clarity. We apologize for the lack of precise labeling in the original figure, which could lead to misinterpretation.

1).Clarification: Figure 4A is designed to visually support the correlation trend observed within the meningioma patient (MP) cohort. It displays representative tumor tissue images from two individual MP patients with contrasting, yet representative, profiles of *Escherichia_Shigella* abundance and corresponding immune infiltration:

Top row (Patient MP2): Exhibits moderate intratumoral immune cell density, corresponding to a moderate fecal relative abundance of *Escherichia_Shigella* (0.103).

Bottom row (Patient MP13): Exhibits higher intratumoral immune cell density, corresponding to a higher fecal relative abundance of *Escherichia_Shigella* (0.556).

2). Modifications Made: To ensure immediate and unambiguous interpretability, we have revised Figure 4A as follows:

Added direct sample identifiers on the figure, labeling the top row as “MP2” and the bottom row as “MP13” .

Updated the figure legend to explicitly state: “ (A) Representative immunohistochemical images of MPO⁺ neutrophils, CD68⁺ macrophages, and CD3⁺ T cells in meningioma tissues from two individual patients (MP2 and MP13) with distinct fecal *Escherichia_Shigella* relative abundance (0.103 and 0.556, respectively).

Scale bar = 50 μm.”

We believe these specific annotations now provide clear, sample-level context that directly illustrates the correlative trend analyzed in panels B. We appreciate your suggestion, which has significantly improved the figure's clarity and interpretability.

Re: Spectrum02485-25R1 (**Gut Microbiota Composition and Tumor Immune Features in Meningioma Patients**)

Dear Dr. Fang Wang:

Your manuscript has been accepted, and I am forwarding it to the ASM production staff for publication. Your paper will first be checked to make sure all elements meet the technical requirements. ASM staff will contact you if anything needs to be revised before copyediting and production can begin. Otherwise, you will be notified when your proofs are ready to be viewed.

Sincerely,
Francesca Benedetti
Editor
Microbiology Spectrum